# Innovations in Drug Discovery for Sickle Cell Disease Targeting Oxidative Stress and NRF2 Activation—A Short Review

**DOI:** 10.3390/ijms26094192

**Published:** 2025-04-28

**Authors:** Athena Starlard-Davenport, Chithra D. Palani, Xingguo Zhu, Betty S. Pace

**Affiliations:** 1Department of Genetics, Genomics and Informatics, College of Medicine, University of Tennessee Health Science Center, Memphis, TN 38103, USA; astarlar@uthsc.edu; 2Department of Pediatrics, Division of Hematology/Oncology, Augusta University, Augusta, GA 30912, USA; cpalani@augusta.edu (C.D.P.); xzhu@augusta.edu (X.Z.); 3Georgia Cancer Center, Augusta University, Augusta, GA 30912, USA; 4Department of Molecular and Cell Biology, Augusta University, Augusta, GA 30912, USA

**Keywords:** sickle cell disease, oxidative stress, NRF2, glutathione, heme, hemolysis, fetal hemoglobin, hemoglobin S, reactive oxygen species

## Abstract

Sickle cell disease (SCD) is a monogenic blood disorder characterized by abnormal hemoglobin S production, which polymerizes under hypoxia conditions to produce chronic red blood cell hemolysis, widespread organ damage, and vasculopathy. As a result of vaso-occlusion and ischemia-reperfusion injury, individuals with SCD have recurrent pain episodes, infection, pulmonary disease, and fall victim to early death. Oxidative stress due to chronic hemolysis and the release of hemoglobin and free heme is a key driver of the clinical manifestations of SCD. The net result is the generation of reactive oxygen species that consume nitric oxide and overwhelm the antioxidant system due to a reduction in enzymes such as superoxide dismutase and glutathione peroxidase. The primary mechanism for handling cellular oxidative stress is the activation of antioxidant proteins by the transcription factor NRF2, a promising target for treatment development, given the significant role of oxidative stress in the clinical severity of SCD. In this review, we discuss the role of oxidative stress in health and the clinical complications of SCD, and the potential of NRF2 as a treatment target, offering hope for developing effective therapies for SCD. This task requires our collective dedication and focus.

## 1. Introduction

Sickle cell disease (SCD) is a global health problem affecting millions worldwide, in sub-Saharan Africa, where 4–6 million people live with the disease [1,2]. The urgency of this problem is underscored by the fact that 90% of the 150,000 babies born annually in Nigeria will die by 5 years of age [3]. It is imperative that we change these statistics by developing robust newborn screening programs and designing safe and effective low-cost oral agents to treat children affected with SCD.

The most common genetic mutation causing SCD is the A to T transversion in the *HBB* gene on chromosome 11, leading to a substitution of valine for glutamic acid in the sixth codon [4] of the β-globin protein chain and the production of sickle hemoglobin S (HbS). Due to the process of hemoglobin switching during the first year of life when the γ-globin to β-globin switch occurs [5], the manifestations of SCD are delayed. However, as fetal hemoglobin (HbF) decreases and HbS levels increase, the pathophysiology of SCD is observed clinically. Under hypoxia conditions, HbS forms polymers, thus producing sickled red blood cells (RBCs), vaso-occlusive episodes, and tissue damage due to ischemia [6]. Inheritance of the homozygous hemoglobin SS genotype produces sickle cell anemia, the most common subtype of the hemoglobinopathies observed worldwide. The common forms of SCD, include HbSC, HbSβ^0^-Thalassemia, HbSβ^+^-Thalassemia, and many others, comprise this group of disorders [7].

The clinical manifestations of SCD consist of chronic hemolytic anemia and recurrent acute vaso-occlusive pain episodes that contribute to clinical severity, morbidity, and early mortality [7]. Common complications include splenic infarcts, high infection rates, dactylitis, stroke, acute chest syndrome, and pulmonary hypertension, among others. The kidneys are particularly susceptible to vaso-occlusive episodes due to functional abnormalities produced by RBCs sickling in the hypoxic microenvironment, producing focal and segmental glomerular sclerosis [8]. As more individuals with SCD survive to adulthood, the prevalence of chronic kidney disease is increasing by 15–30%, with significant mortality due to end-stage kidney disease and a lack of tissue-matched transplant donors in the US.

In addition to the formation of sickled RBCs by HbS polymerization, it also triggers hemolysis and the release of Hb and free heme, causing inflammation, oxidative stress, and vascular–endothelial dysfunction [9,10]. Excessive free radicals produced by the release of Hb cause the activation of pro-oxidant enzymes and the release of reactive oxygen species (ROS), mediating oxidative stress [2,11]. The primary function of mitochondria is the process of oxidative phosphorylation, which controls ROS production and ATP synthesis to provide cells with energy. Although normal RBCs lose mitochondria when fully mature, that is not the case for individuals with SCD. Rivers et al. [12,13] demonstrated abnormal mitochondrial retention in people and mice with SCD, producing ROS-associated hemolysis. Subsequent work from the Rivers lab showed that stress erythropoiesis contributes to mitochondrial retention and ROS generation in SCD [14].

Emerging evidence supports a pivotal role in mitochondrial dysfunction in amplifying oxidative stress in SCD. Mitochondria both creates and neutralize ROS generated from different sources, and its impaired homeostasis observed in SCD exacerbates oxidative damage. Elevated mitochondrial ROS, coupled with a compromised antioxidant capacity such as low NRF2 (Nuclear factor erythroid 2-related factor 2) and superoxide dismutase 2 levels, contribute to endothelial injury, inflammation, and hemolysis. Targeting the mitochondrial axis—through strategies to improve function or promote mitophagy—offers promising therapeutic approaches to restore redox balance and ameliorate the clinical severity of SCD.

A perfect target under these circumstances is the transcription factor NRF2, which is the master regulator of the antioxidant response in cells [15,16]. This factor is sequestered in the cytoplasm by Kelch-like ECH-associated protein 1 (KEAP1) and β-transducin repeat-containing protein and directed to the proteasome for degradation. However, under oxidative stress conditions, NRF2 is released and translocates to the nucleus where it activates several antioxidant proteins, including heme oxygenase 1 (HMOX1), quinone oxidoreductase 1, glutamate-cysteine ligase catalytic subunit, and glutamate-cysteine ligase modifier subunit. These proteins work together to reduce cellular ROS and inflammation. Our group and others have demonstrated another role of NRF2 as a modulator of γ-globin gene transcription and HbF expression [17,18,19]. In the case of SCD, enhancing NRF2 levels provides a unique benefit through HbF induction, which inhibits HbS polymerization and lowers oxidative stress to ameliorate clinical symptoms. Therefore, US FDA-approved oral activators of NRF2, such as dimethyl fumarate (DMF) [20,21] and simvastatin [22,23], are attractive targets for drug repurposing in SCD.

Efforts to target oxidative stress using small oral-active chemical molecules to decrease the toxic side effects are widespread. However, despite these efforts, there remains a paucity of drugs approved for the clinical treatment of oxidative stress in SCD. Hydroxyurea (HU) was the first US FDA-approved agent specifically for treating SCD, mediating several clinical benefits, including HbF induction, which blocks HbS polymerization, decreasing inflammation and oxidative stress, and increasing nitric oxide production [24,25]. These benefits contribute to the amelioration of pain episodes and other complications in adults and children with SCD. However, there remain concerns over the long-term use of HU and its effects on fertility and secondary malignancy; the latter has not occurred in SCD after more than 25 years of clinical use [26,27]. In the REACH trial, children living in sub-Saharan Africa received HU at a fixed dose or with dose escalation. After an adequate evaluation period, the data safety monitoring board halted the trial when the number of clinical events was significantly lower among children receiving escalated dosing of HU when compared to a fixed dose [27]. They had fewer vaso-occlusive pain episodes, acute chest syndrome, transfusions, and hospitalizations. Laboratory tests confirmed similar toxicity in the two groups. However, there were no cases of severe neutropenia or thrombocytopenia.

Over the last 10 years, three additional agents were US FDA-approved for the treatment of SCD, including l-glutamine [28] and crizanlizumab [29,30], which indirectly target oxidative stress and voxeletor [31], which inhibits HbS polymerization to improve total hemoglobin levels. Recently, during the fall of 2024, there was an abrupt discontinuation of voxelotor due to safety concerns. In the meantime, extraordinary progress was made by pharma companies in 2023 with the US FDA approval of the first two gene therapies for SCD, Casgevy (exagamglogene autotemcel) [32,33] and Lyfgenia (lovotibeglogene autotemcel) [34,35]. Whether these therapies will be curative requires long-term follow-up. However, there was a remarkable resolution of vaso-occlusive episodes, marker of chronic hemolysis, and oxidative stress levels, along with improved quality of life for individuals treated with both innovative drugs. Whether there will be equal accessibility to gene therapy for individuals with SCD in the US or globally is a challenging question for healthcare providers to answer.

In this review, we will discuss progress made in the field to understand the role of oxidative stress in human health and the pathophysiology of SCD. We will also examine current strategies for developing small molecules and NRF2-activating drugs for treating oxidative stress. Specifically, we will explore the impact of these therapies in the field, highlighting the promising role of targeting NRF2 to improve clinical outcomes and reducing the global burden of SCD.

## 2. Oxidative Stress

The pathological state of oxidative stress arises from an imbalance between the production of ROS and the capacity of the antioxidant defense system to neutralize it. Under physiological conditions, ROS plays an essential role in cellular signaling and homeostasis [36]. Various natural biological processes in the human body, including respiration, digestion, alcohol and drug metabolism, and the conversion of fats into energy, generate ROS as a metabolic byproduct [37]. Moderate levels of ROS are beneficial for protecting against pathogens and wound healing activities [38]. However, excessive ROS accumulation leads to cellular and tissue damage through lipid peroxidation, protein oxidation, and DNA damage, contributing to disease pathogenesis [37,39,40,41].

Reactive oxygen species are generated from various sources in the human body, such as the phase I cytochrome P450 metabolizing enzymes, NADPH oxidases, and the mitochondrial electron transport chain (ETC). The P450 system utilizes molecular oxygen to form superoxide anions and other ROS [42], which participate in cell signaling or contribute to oxidative stress and tissue damage [42,43]. Membrane-bound enzymes like NADPH oxidases generate superoxide in response to cell signaling activated by inflammatory stimuli. These enzymes play a crucial role in immune defenses by producing ROS to kill pathogens; however, sustained oxidative damage causes chronic disease states [44]. Other enzymatic sources of ROS include xanthine oxidase during purine metabolism and uncoupled endothelial nitric oxide synthase generating superoxide under conditions of endothelial dysfunction [45,46]. However, the primary generator of oxidative stress is mitochondria through the ETC [47]. These mechanisms regulate ROS levels to prevent oxidative stress and tissue damage [48,49]. As a physiological mechanism, NRF2, superoxide dismutase, catalase, and glutathione peroxidase are the primary antioxidants to neutralize oxidative stress [50].

## 3. Electron Transport Chain

The ETC in mitochondria is crucial for energy production in eukaryotic cells. It primarily generates ATP while being a significant source of oxidative stress (Figure 1). Mitochondria utilize about 90% of the body’s oxygen for ATP production through oxidative phosphorylation, leading to ROS generation as a byproduct [51,52]. Under normal conditions, the NRF2-mediated antioxidant defense system neutralizes excess ROS, but when oxidative stress exceeds its capacity to neutralize, irreversible cellular damage occurs [48,49].

Glucose is a primary energy source, undergoing glycolysis in the cytoplasm to yield pyruvate, which is then oxidized to acetyl-CoA or carboxylated to oxaloacetate before entering the tricarboxylic acid cycle, generating NADH and FADH₂ for controlling ROS [53]. The ETC located in the inner mitochondrial membrane consists of four protein complexes, with ubiquinone and cytochrome c as mobile electron carriers. A series of electron transfers through Complexes I-IV reduces molecular oxygen to water (Figure 1) to create a proton gradient utilized for ATP synthesis [54,55].

Antioxidant enzymes such as superoxide dismutase, catalase, and glutathione peroxidase are vital for mitigating oxidative stress by oxidative phosphorylation [56]. Superoxide dismutase converts superoxide radicals into oxygen and hydrogen peroxide to protect cellular structures. Mitochondrial superoxide dismutase 2 is crucial for detoxifying ROS; deficiencies can worsen oxidative damage and be linked to disease manifestations [57,58,59]. Catalase metabolizes hydrogen peroxide into water and oxygen and is critical for protecting cells from oxidative injury in ROS-producing tissues such as the liver [60]. Glutathione peroxidase reduces hydrogen peroxide using glutathione to maintain cellular homeostasis and host defenses against oxidative damage in diseases characterized by high oxidant stress [61,62].

## 4. Oxidative Stress in Sickle Cell Disease

Oxidative stress is produced by unique mechanisms in SCD. First, the level of hypoxia present in postcapillary venules triggers the polymerization of HbS, leading to the formation of sickled RBCs that undergo autoxidation which generates excess ROS (Figure 1). Subsequently, oxidative stress damages the RBC membrane and activates lipid peroxides, accelerating cellular breakdown within blood vessels—a process known as intravascular hemolysis. The net result is the release of free HbS and heme from RBCs, which is accompanied by the formation of microparticles and the release of damage-associated molecular patterns, such as Hsp-70 and interleukin-33, accelerating the generation of ROS, thus perpetuating the cycle of oxidative damage.

Both microparticles and damage-associated molecular patterns increase blood viscosity and stimulate the formation of inflammasomes by interacting with macrophages through toll-like receptor 4 (TLR-4) binding and the activation of the NF-kB and NLRP3 signaling pathways. These activated immune cells and sickled RBCs recruit endothelial cells to expand the inflammatory niche initiating vaso-occlusion, vasculopathy, and end organ ischemic-reperfusion injury [18,63]. This toxic process contributes to cardiac dysfunction, leading to cardiomyopathy, heart failure, and pulmonary hypertension because of repeated injury to cells [64,65,66,67], among other complications.

### Environmental Factors

Parallel with endogenous ROS production, environmental stressors, including infections and aging, contribute to oxidative stress in SCD. For example, pneumonia causes an inflammatory response and triggers immune activation, generating ROS by neutrophils and macrophages. These cells produce oxidative bursts and secrete effector proteins or toxins that interfere with the translocation of the NADPH oxidase complex, exacerbating endothelial dysfunction and RBC hemolysis [68,69]. Additionally, aging is associated with mitochondrial dysfunction, reduced antioxidant enzyme activity, and cumulative oxidative damage [70], all of which worsen the oxidative burden in SCD [71], underscoring the need for targeted antioxidant therapies for clinical management [69].

The process of vaso-occlusion and oxidative stress affects all body organ systems, which is especially detrimental in SCD, where chronic oxidative stress exists. For example, cardiopulmonary complications are the leading cause of death due to diastolic heart failure and pulmonary hypertension [72]. Chronic hemolysis produces excessive pro-oxidant enzymes, free HbS and heme, which promotes hydroxy radicals accumulation via the Fenton reaction [2], and the retention of mitochondrial ETC activity [73,74,75].

Splenic sequestration crisis is another major complication of SCD mediated by oxidative stress that occurs when sickled RBC obstructs the draining vein in the red pulp of the spleen. This leads to the enlargement of this organ [76] and disordered architecture of the white pulp. Over time, the number of splenic follicles are reduced and replaced by fibrotic tissue from RBC congestion in red pulp [77], and eventual ischemic necrosis and infarction, i.e., autosplenectomy [78,79].

The scale of liver disease in SCD ranges from hepatocyte damage to severe liver failure associated with multiple organ failure syndrome [80,81], accounting for 3–11% of deaths in SCD [82,83,84]. Oxidative stress related to iron overload markedly increases malondialdehyde, a marker of lipid peroxidation [85], and high levels of myeloperoxidase released by activated leukocytes and neutrophils, causing endothelial dysfunction and liver injury [86]. Lastly, acute RBC intrahepatic cholestasis can occur due to sickling in the hepatic circulation, leading to gallstones, ischemia, and liver failure [87].

Oxidative stress-mediated endothelial dysfunction and ischemia-reperfusion injury play a key role in developing proliferative retinopathy [88], which causes vision loss and blindness in SCD. This complication affects mainly in adults due to retinal ischemia with secondary neovascularization and hemorrhage, mediating retinal artery infarction and retinal detachment [89]. Reactive oxygen species cause endothelial cell damage through oxidative reactions with membrane lipids, peptides, and nucleic acids [90,91], exposing subendothelial structures and proteins, including tissue factor, causing a hypercoagulable state [92]. Previous studies demonstrated improvement in retinal pigment epithelial cells after treatment with the oral antioxidant monomethylfumarate in SCD mice [93].

Finally, sickle nephropathy is a significant complication of SCD, with up to 18% of affected individuals developing end-stage renal disease [94,95,96]. Oxidative stress occurs in the kidney due to high heme content and lipid peroxidation [97,98,99] producing malondialdehyde and damage to the cell membrane that plays a key role in the pathophysiology of sickle nephropathy [63]. Oxidative stress damages the endothelial lining of kidney blood vessels, leading to proteinuria, impaired glomerular filtration, hypertension, and the development of chronic kidney disease and organ failure [99,100,101].

## 5. Murine Studies Linking Oxidative Stress to SCD Clinical Severity

Several published murine studies in the Townes SCD mouse model for drug development show the involvement of oxidative stress in the severity of the phenotype. Simvastatin (Zocor) is a statin drug developed to decrease cholesterol levels in humans by inhibiting the function of hydroxymethylglutaryl-coenzyme A reductase [102,103]. This agent modulates ROS levels and activates NRF2 expression and PI3K/AKT signaling. Simvastatin controls iron metabolism through *HMOX1* activation [104] and glutathione reduction in the liver [105], lungs [106], and spleen. Our group recently published data generated in SCD mice treated with simvastatin (Table 1) showing the reversal of HbS sickling and a decrease in ROS stress levels [23]. A phase I/II clinical trial with short-term simvastatin treatment in SCD patients showed increased nitric oxide levels and decreased C-reactive protein and interleukin-6 expression. However, no effects on vascular endothelial growth factor, vascular cell adhesion molecule-1, or tissue factor protein levels were observed [22].

Our group also conducted studies in SCD mice treated with another NRF2 activator dimethyl fumarate (DMF), which inhibits cytoplasmic sequestration of NRF2 by KEAP1 leading to nuclear translocation and activation of its downstream target gene *HMOX1,* and *NQO1* among others (Figure 2) [21]. DMF also reduces inflammation and organ failure and improves the SCD phenotype [18,122], supported by decreased proinflammatory cytokines, such as interleukin-6 and interleukin-1β, and heme, and oxidative stress levels. Moreover, our group showed that DMF induces HbF expression and reduced the percentage of sickle RBCs under hypoxia conditions [21]. On the contrary, in a SCD NRF2 knockout mouse, we showed that *HMOX1* and *HBG* gene expression were silenced [112], and disease severity increased significantly [122]. These findings support the rationale for activating NRF2 as a therapeutic approach to induce HbF expression, reduce oxidative stress, and improve organ function [112].

Various other compounds have been evaluated in SCD mice as NRF2 activators. For example, CDDO-Im (1-(2-cyano-3,12,28-trioxooleana-1,9(11)-dien-28-yl)-1H-imidazole) reduces inflammation and improves organ function in SCD mice [112]. Similarly, when SCD mice were treated with 3H-1,2-dithiole-3-thione, a small chemical inducer of NRF2, a decrease in acute chest syndrome and increased survival were observed [123]. In SCD mice, activated neutrophils release myeloperoxidase, which generates oxidants that impair endothelial cell function. Zhang et al. showed that inhibiting myeloperoxidase decreases myeloperoxidase-dependent oxidative stress and restores endothelial cell function [86]. The inhibition of NADPH oxidase decreases oxidative stress to improve vascular function in SCD [124]. Chronic hemolysis produces elevated plasma free hemoglobin and heme, which activates TLR4-mediating ROS generation. Of note, the treatment of BERK SCD mice microglial cells with TLR4 inhibitors reversed this effect [125].

Curcumin is a polyphenol chemical belonging to the group of curcuminoids responsible for the yellow color in turmeric. Studies performed with intestinal HCT-116 and HT-29 cell lines demonstrated the ability of curcumin to reduce oxidative stress and ROS levels and restore mitochondrial transmembrane potential through the action of superoxide dismutase, catalase, and glutathione [126]. Curcumin also mediates NRF2 activation, binding to the antioxidant response elements of downstream antioxidant genes involved in cell survival. Although curcumin is safe, due to limited oral bioavailability, clinical development has been hampered. More recently, a novel transdermal curcumin gel was shown to ameliorate pain hypersensitivity and reduce RBC hemolysis in a SCD mouse, thus providing the impetus for continued development [114].

Salubrinal is another small molecule that functions as a selective eIF2α dephosphorylation inhibitor, primarily used to study stress responses in eukaryotic cells. It was initially shown to increase the number of translating ribosomes on γ-globin mRNA to enhance translation efficiency [110]. In vitro studies from our group with salubrinal, demonstrated HbF induction through p-eIF2α and ATF4 activation in the stress-signaling pathway [115]. We also observed HbF induction in sickle erythroid progenitors treated with salubrinal and a reduction in ROS. Chronic treatments of SCD mice with salubrinal mediated a significant increase in HbF expression and reduced the percentage of sickled RBCs in peripheral blood.

## 6. Targeted Therapy to Reduce Oxidative Stress in SCD

Hydroxyurea is the only FDA-approved drug in the United States recommended as the standard of care for treatment of adults and children with SCD. Its main mechanism of action is HbF induction, which inhibits HbS polymerization, the formation of sickle RBCs, and vaso-occlusive episodes. Long-term, HU treatment also improves hemoglobin and nitric oxide levels, produces an anti-inflammatory response, and improves survival rates of individuals with SCD. With this established recommendation, the majority of clinical trials conducted in the last decade continue HU treatment at an established maintenance dose while testing new agents. This approach indirectly addresses the ability to develop additive or synergistic drug regimens comprising agents with different mechanisms of action for individuals with SCD. For example, HU combined with DMF produces an additive effect on HbF induction in vitro without intolerable toxicity. As more oral antioxidant and HbF-inducing drugs are evaluated in clinical trials and shown to be effective and safe, combination treatment regimens will become a reality for individuals with SCD; potential novel drug targets are discussed below.

### 6.1. N-Acetyl Cysteine (NAC)

N-acetyl cysteine (NAC) provides cysteine for the de novo biosynthesis of glutathione and has been investigated for its effects on cellular oxidative damage, coagulation, and endothelial activation. We were the first to show in an early phase I trial that NAC inhibits dense and irreversible sickle RBC formation and restores glutathione levels toward normal in adolescents and young adults with SCD [127]. In addition, we observed a trend towards resolution of vaso-occlusive episodes after a 6-month treatment period with NAC only; subsequently, other groups confirmed our findings. Nur et al. later demonstrated that 6 weeks of NAC treatment increases whole blood glutathione levels and decreases plasma advanced glycation end-products and cell-free hemoglobin levels (Table 2); moreover, none of the SCD patients experienced painful episodes or other significant NAC-related complications while undergoing treatment [128]. In addition, Fu et al. demonstrated that NAC increases total and free-thiol concentrations of cysteine and glutathione after intravenous administration, thus relieving oxidative stress; however, only one SCD patient was evaluated in their study [129].

Previous clinical trials demonstrated the beneficial effects of glutamine in preventing SCD vaso-occlusive crises [28]. SCD patients treated with l-glutamine showed significant increases in NADH and NAD redox potential. Furthermore, l-glutamine decreased endothelial adhesion by sickle RBCs in vitro, suggesting that clinical improvement occurred through oxidative stress mechanisms. The ability of NRF2 to regulate glutamine metabolism is another reason to support the development of this small molecule activator for treating SCD.

### 6.2. Hemopexin

Hemopexin is a crucial plasma protein that binds and transports heme for scavenging and levels are significantly reduced in individuals with SCD. Hemopexin was shown to prevent heme-iron loading in the cardiovascular system, thus limiting the production of ROS and the activation of adhesion molecules; hemopexin also promoted heme recovery and detoxification by the liver mainly through the activation of *HMOX1* [134,135]. In SCD mice, treated three times a week for three months, hemopexin produced a dose-dependent reduction in heme exposure and pulmonary hypertension while improving cardiac pressure–volume relationships and exercise tolerance. In addition, hemopexin attenuated pulmonary fibrosis and oxidative modifications in the lung and myocardium of the right ventricle [136]. More widely, hemopexin reduces endothelial exposure of P-selectin and von Willebrand factor, stimulating hepatic *HMOX1* to decrease vascular inflammation [137]. These studies demonstrate preclinical therapeutic proof-of-concept for the development of hemopexin for treating SCD.

Recently, plasma-derived human hemopexin CSL889 was investigated in a phase 1 study to evaluate its safety, tolerability, and pharmacokinetics in SCD patients [138]. The results showed that CSL889 has an excellent safety and tolerability profile when administered as a single dose up to 200 mg/kg in subjects in steady-state and at 60 mg/kg in subjects in an acute vaso-occlusive episode (Table 2). These results provide a strong foundation for future trials to evaluate the efficacy of hemopexin in SCD [138].

### 6.3. Alpha-Lipoic Acid

Alpha-lipoic acid (ALA) is another potent antioxidant drug that has been investigated in the treatment of several diseases. Recently, Simonia et al. reported that ALA reduced platelet activation and thrombus formation in SCD mice [139]. High dietary intake of ALA reduced sickle RBCs, liver fibrosis, and adhesion molecule expression, in multiple organs as well. Martins et al. evaluated the effects of ALA in sickle trait and SCD individuals. Unfortunately, ALA (200 mg) decreased glutathione peroxidase 4 activity in both groups while increasing catalase activity and reducing lipid and protein damage only in sickle trait subjects [131]. These results indicate that more studies are required to optimize the dose of ALA to produce therapeutic efficacy in SCD.

### 6.4. δ-Aminolevulinate

Previously, we investigated the ability of δ-aminolevulinate, the heme precursor, to activate *HBG* expression and its effects on cellular functions in erythroid cell systems. We demonstrated that this small molecule induced *HBG* transcription and γ-globin protein chain synthesis in KU812 erythroid cells [132]. Using inhibitors of the heme biosynthesis pathway, we demonstrated that heme participated in δ-aminolevulinate-mediated γ-globin protein expression via NRF2 activation. These data support future studies to explore the potential of stimulating intracellular heme biosynthesis by δ-aminolevulinate as a novel therapeutic strategy for NRF2 activation in SCD.

### 6.5. L-Arginine

Arginine levels are acutely deficient in SCD patients due to consumption during vaso-occlusive pain episodes. When levels are low, superoxide is produced instead of nitric oxide, reducing its bioavailability and generating oxidative stress. Arginine recently emerged as a new treatment option for SCD since intravenous treatment was shown to improve mitochondrial function and reduce oxidative stress in children and young adults with SCD during vaso-occlusive pain episodes (Figure 2) [140]. A phase 3 randomized controlled trial, Sickle Cell Disease Treatment with Arginine Therapy (STArT), will test the efficacy of arginine treatment in 360 children, adolescents, and young adults in an acute vaso-occlusive pain episode [141]; the STArT trial is in progress.

### 6.6. Sulforaphane

Sulforaphane is a sulfur-rich member of the isothiocyanate family of compounds found in cruciferous vegetables. Previous genomic work demonstrated that the reduction in oxidative stress capacity of SCD may be due to low NRF2 levels [107]. In vitro studies showed the ability of sulforaphane to activate NRF2 and downstream target genes, *HMOX1*, *NQO1*, and *HBG* in erythroid progenitors. Subsequently, a phase 1 study of sulforaphane in a broccoli sprout homogenate was conducted in adults with SCD. They observed an increase in *HMOX1* levels and a trend toward increasing *HBG* mRNA but no significant change in *NRF2* transcription or HbF protein levels [133]. Another group took a genetic approach to inhibit Keap1 in SCD mice, which mediated NRF2 induction. SCD mice were also treated with sulforaphane where they observed an increase in NRF2-dependent cytoprotective genes, amelioration of liver damage, and reduced free heme levels [142].

## 7. Role of NRF2 in Globin Gene Regulation

The reactivation of HbF expression is a key therapeutic option in developing novel therapies for SCD. It is well established that HbF protein inhibits the polymerization of HbS to reverse the pathophysiology of SCD and disease severity. The FDA-approved drug HU and other small molecule compounds, such as butyrate and decitabine, were shown to induce HbF expression, which inhibits HbS polymerization under hypoxic conditions [143,144,145,146]. Indeed, multiple transcription factors participate in the regulation of the five functional globin genes located in the *HBB* locus on chromosome 11. Interestingly, HU has been shown to remodel the *HBG* promoter in association with GATA1, GATA2, NFY, and BCL11A proteins to facilitate an open chromatin structure [147], while butyrate and decitabine affect *HBG* transcription through modifications of proximal promoter histone acetylation and DNA methylation levels, respectively [144,145,146].

Additionally, CRISPR strategies showed significantly higher *HBG* expression in erythroid cells carrying mutations producing hereditary persistence of HbF which reverse the phenotypes of SCD and β-thalassemia major [148,149,150]. An important finding by creating insertions and deletions of the binding sites for the repressors, BCL11A and Leukemia/lymphoma-related factor in the proximal *HBG* promoter, resulted in gene reactivation and reversal of the sickle RBC phenotype [151,152]. The corresponding mechanistic studies support the notion that HbF induction is associated with epigenetic modifications and changes in chromatin contacts within the *HBB* locus.

The critical transcription factor NRF2 was originally identified as a DNA-binding protein in the *HBB* locus and later was characterized as the major regulator of oxidative stress [111]. Previous studies conducted by Lowrey and colleagues demonstrated enhanced NRF2 binding in the *HBG* promoter and HbF induction after tert-butylhydroquinone and simvastatin treatment [17,109]. Likewise, our group investigated the ability of DMF to activate HbF expression in primary human erythroid progenitors through NRF2 binding in the *HBG* antioxidant response element [18]. Using JASPAR61 software (http://jaspar2016.genereg.net/; accessed 5 December 2016) we identified 23 NRF2 consensus motifs 5′-TGAnnnnGC-3′ in the *HBB* locus. Subsequent studies demonstrated in vivo binding in the locus control region and proximal *HBG* promoters, supporting long-range chromatin looping to regulate globin gene expression during drug treatment [19]. Protein–protein interaction studies demonstrated that NRF2 dimerizes with small MAF proteins to bind DNA and activate gene transcription (Figure 2). Moreover, NRF2 competes with the transcription factor BACH1 (tBTB Domain and CNC Homolog 1), a cap ‘n’ collar protein family member, which competitively binds the antioxidant response elements of target genes to regulate cellular oxidative stress levels [116].

### 7.1. BACH 1 Inhibitors

Mechanistically, heme binds to the transcription factor BACH1 to mediate repression of NRF2-mediated gene transcription. ASP8731 is a selective small molecule inhibitor of BACH1 [117], which increases *HMOX1* transcription in HepG2 liver cells. Likewise, treatment of Townes SCD mice with ASP8731 inhibited heme-mediated microvascular stasis, activated *HMOX1,* and decreased hepatic NF-kB phospho-p65 protein expression. ASP8731 increased *HBG* transcription and F-cells percentages. In human erythroid progenitors generated from CD34^+^ stem cells, ASP8731 increased *HBG* mRNA levels and the percentage of F-cells 2-fold. These data indicate that BACH1 inhibitors may offer a new therapeutic target to activate HbF expression for treating SCD.

Another novel BACH1 inhibitor, HPPD, was developed by Attucks and colleagues at vTv Therapeutics. They showed that HPPD promotes the nuclear export of BACH1 and the translocation of NRF2 to activate *HMOX1* expression [118]. In a subsequent study, we showed the ability of HPPD to attenuate BACH1 and enhance NRF2 protein levels. This was followed by ChIP assays demonstrating NRF2 binding in the *HBG* antioxidant response element and *HBB* locus control hypersensitive site 2 in KU812 erythroid cells; in addition, HPPD activated *HBG* gene transcription and HbF protein synthesis [153]. To validate the potential of HPPD for treating SCD, an anti-sickling study was conducted in primary erythroid cells generated from individuals with SCD. Cells incubated with HPPD under hypoxia conditions showed a 50% reduction in the number of sickled RBC, supporting HbF induction by HPPD and the mediation of an anti-sickling phenotype in SCD erythroid progenitors.

### 7.2. Regulation by MicroRNA Genes

Recently, miR-144 and miR-451 (miR-144/451) were demonstrated to play a significant role in erythroid differentiation and fine-tuning gene expression during erythropoiesis [119,120]. Interestingly, in adults with SCD, higher miR-144 expression decreased NRF2 and glutathione levels and was associated with severe anemia [121]. These findings were linked with a lack of the antioxidant proteins glutamate cysteine ligase C/M and superoxide dismutase 1 [154]. To further investigate this observation, we conducted genome-wide microarray analysis to discover miRNA genes associated with high and low HbF expression in individuals with SCD. We observed the upregulation of miR-144 in subjects with lower HbF levels than those with higher HbF levels [121]. Functional studies in primary adult erythroid progenitors showed *NRF2* silencing by miR-144 and repression of *HBG* transcription; by contrast, treatment with an miR-144 antagomir reversed its silencing effects on *NRF2* expression. Additional studies have shown miR-28 inhibits *NRF2* through a KEAP1-independent mechanism and, conversely, miR-153, miR-27a, and miR-142-5p down-regulate *NRF2* expression [155,156,157]. These studies support miRNA-mediated mechanisms of *NRF2* regulation as potential for *HBG* activation.

## 8. Future Directions

Targeting oxidative stress through *NRF2* activation represents a promising future strategy for discovering drugs to treat SCD, since this transcription factor plays a pivotal role in cellular defenses against oxidative stress through the enhanced transcription of antioxidant genes. As the ischemia-reperfusion injury and inflammatory characteristic of SCD lead to increased ROS formation, enhancing NRF2 protein activity should mitigate these effects and improve clinical outcomes. However, the translation of *NRF2*-targeted therapies from preclinical studies to clinical application remains limited. This gap is due to funding limitations for clinical trials and the number of investigators with the expertise to lead these endeavors to rigorously evaluate the safety and efficacy of *NRF2* activators in SCD.

Inter-individual variability poses another clinical challenge in harnessing *NRF2* activators and antioxidant small molecules as treatment strategies for SCD. Genetic factors, such as polymorphisms in *NRF2* or related antioxidant genes, can influence steady-state oxidative stress levels and the response to activators. Additionally, environmental exposures, dietary habits, and lifestyle choices significantly modulate oxidative stress responses, leading to variability in treatment outcomes. Understanding these influences is crucial to tailoring *NRF2*-targeted therapies and developing personalized treatment plans. Future studies should focus on characterizing these factors in diverse populations to maximize the efficacy of *NRF2*-targeted interventions in managing oxidative stress associated with SCD.

## 9. Conclusions

Sickle cell disease is a common genetic disorder caused by a single-point mutation in the *HBB* gene, leading to HbS synthesis and polymerization under hypoxic conditions. The net result is a chronic intravascular hemolytic anemia, producing a severe oxidative stress state due to the release of hemoglobin and free heme into the plasma. Coupled with RBCs sickling, vaso-occlusion, and ischemia-reperfusion damage to internal organs, individuals with SCD suffer high morbidity and mortality. Therefore, reducing oxidative stress and enhancing HbF levels is an efficacious strategy to develop novel therapies for SCD.

As we move into the era where we have multiple FDA-approved drugs, we can start to develop combination regimens and take a multi-pronged treatment approach, especially with novel small molecules that act through different molecular and cellular mechanisms. We reviewed the most promising agents that either lower ROS and/or induce HbF to produce a synergistic effect on ameliorating the pathophysiology of SCD. Thus far, the most efficacious clinically tested combination has been HU and voxeletor, but the latter was recently removed from the market. Fortunately, we know that NRF2 is the major regulator of oxidative stress and serves as a trans-activator of the *HBG* promoter, with the potential as a single agent capable of increasing HbF protein and dampening oxidative stress. Several drugs are ripe for the picking that merit clinical safety testing in individuals with SCD, such as the FDA-approved small molecule dimethyl fumarate.

Agents with anti-inflammatory properties are also desirable for inhibiting the release of cytokines that cause tissue damage. Hydroxyurea is an effective therapy for SCD; however, some individuals do not respond, and there are concerns over long-term safety and adverse effects. Although drug development has progressed, there remains a need for expanded preclinical testing in SCD mice to grow the repertoire of drugs available for clinical trials. Lastly, from the perspective of a global lens, we must develop oral drugs that are safe, efficacious, and low-cost for individuals in Africa, India, and South America where the majority of people affected with SCD live.

## Figures and Tables

**Figure 1 ijms-26-04192-f001:**
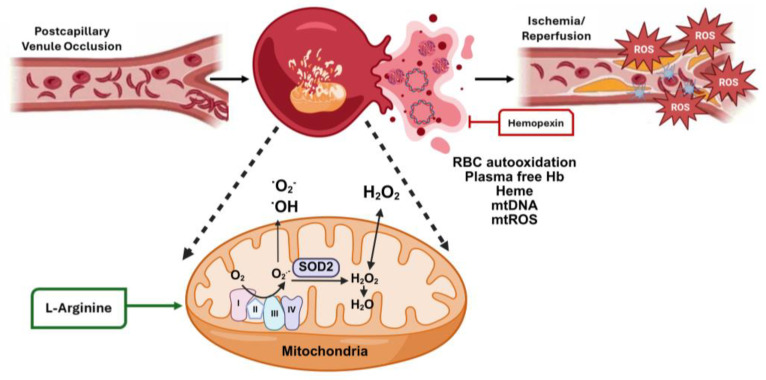
Consequences of chronic hemolysis and mitochondrial dysfunction in SCD. Hemoglobin S (HbS) polymerization leads to red blood cell (RBC) membrane damage and intravascular hemolysis releasing free Hb and heme in the plasma. Over time, the ability of haptoglobin and NRF2 to provide antioxidant protection from excess HbS, heme, and reactive oxygen species (ROS) is overwhelmed, leading to endothelial cell dysfunction, generation of ROS and vaso-occlusion in the post capillary venules under hypoxic conditions. This process eventually produces ischemia/reperfusion organ and tissue damage. Under normal conditions, mitochondria control the generation of ROS in cells, and they are removed by the process of autophagy during RBC maturation. However, chronic hemolysis in SCD leads to the retention of dysfunctional mitochondria that release free radicals (·OH, ·O_2_^−^), hydrogen peroxide (H_2_O_2_), mitochondrial DNA (mtDNA), and ROS (mtROS) to escalate oxidative stress further. Small molecules like hemopexin that directly bind and catabolize free heme and L-arginine reduce plasma heme levels and improve mitochondrial function, respectively. The figure was generated using BioRender (https://app.biorender.com/) accessed 15 April 2025.

**Figure 2 ijms-26-04192-f002:**
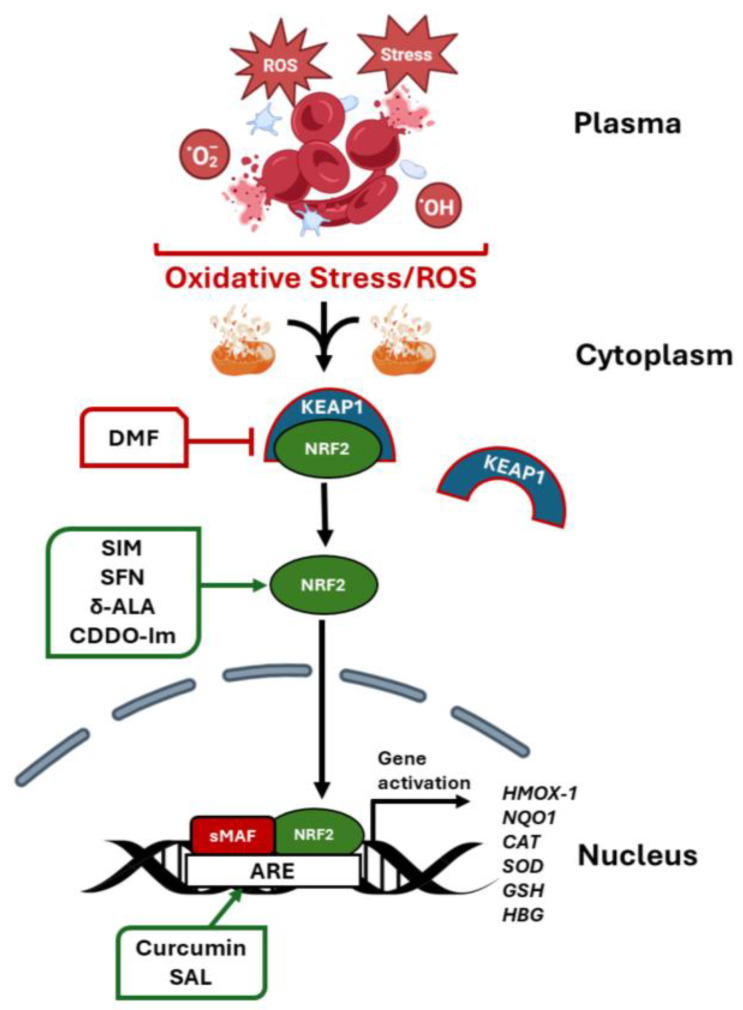
Development of small molecules that activate NRF2 expression, reduce oxidant stress, and induce fetal hemoglobin expression. In SCD, chronic hemolysis is a major contributor to oxidative stress leading to low NRF2 levels [15]. Therefore, small molecules that inactivate KEAP1, the negative cytoplasmic regulator of NRF2, allowing its release and translocation to the nucleus, should be beneficial in diseases with a major oxidative stress component. Once in the nucleus, NRF2 binds the antioxidant response element (ARE) in the proximal promoter of antioxidant genes including *HMOX1* (heme oxygenase 1), *NQO1* (NAD(P)H quinone dehydrogenase 1), *CAT* (catalase), *SOD* (superoxide dismutase), and *GSH* (glutathione). Moreover, NRF2 is a potent activator of the *HBG* genes encoding γ-globin protein chains. Fetal hemoglobin inhibits HbS polymerization to reverse RBC hemolysis and ameliorate the severe SCD clinical phenotype. The novel agents shown support potential therapeutic strategies to develop combination drug regimens with hydroxyurea, the current standard of care for SCD. Abbreviations: δ-ALA, δ-aminolaevulinate; DMF, dimethyl fumarate; SIM, simvastatin; CDDO-Im, 1-(2-cyano-3,12,28-trioxooleana-1,9(11)-dien-28-yl)-1H-imidazole; ROS, reactive oxygen species; SAL, salubrinal; SFN, sulforaphane. The figure was generated using BioRender (https://app.biorender.com/) accessed 15 April 2025).

**Table 1 ijms-26-04192-t001:** Summary of drugs that target NRF2 as a mechanism of γ-globin gene activation.

Drug	Chemical Structure/DNA Sequence	Mechanism of Action	In Vitro/Preclinical SCD Mice Studies	References
Simvastatin	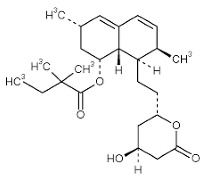	Competitive 3-hydroxy-3-methyl-glutaryl (HMG)-CoA reductase inhibitor NRF2 agonist, decreases ROS levels.	In vitro and preclinical SCD mice studies NCT01702246.	[107,108]
tert-butylhydroquinone	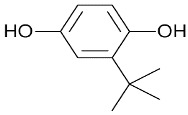	NRF2 agonist mediating glutathione production.	In vitro.	[109]
Dimethyl fumarate	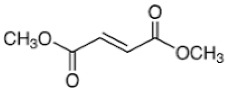	A small-molecule NRF2 agonist, decreases ROS.	In vitro and preclinical SCD mice studies.	[20,21,110,111]
1-(2-cyano-3,12,28-trioxooleana-1,9(11)-dien-28-yl)-1H-imidazole	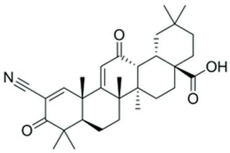	NRF2 agonist and enhanced ARE binding.	NRF2 agonist, decrease ROS and preclinical SCD mice studies.	[110] [107,112]
Curcumin	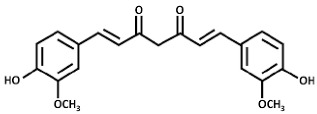	NRF2 agonist and enhanced ARE binding.	In vitro.	[108,113] [114]
Salubrinal	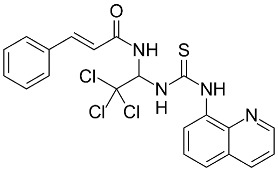	NRF2 agonist and enhanced ARE binding.	In vitro and preclinical SCD mice.	[115]
ASP8731	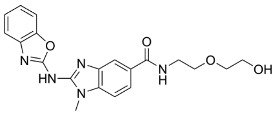	A selective small molecule inhibitor of BACH1.	In vitro and preclinical SCD mice studies.	[116]
HPPD	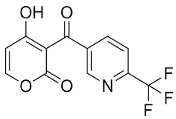	Small molecule inhibitor of BACH1.	In vitro and preclinical SCD mice.	[117,118]
miR-144	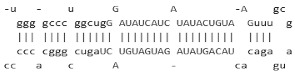	Inhibits NRF2 through miR-144 target sites in the 3′UTR of NRF2.	In vitro.	[119,120,121]

**Table 2 ijms-26-04192-t002:** Summary of drugs that decrease oxidative stress by different mechanisms.

Drug	Chemical Structure	Mechanism of Action	Clinical Trials	References
N-acetyl cysteine	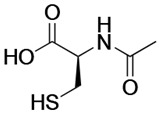	Precursor for glutathione biosynthesis, decreases ROS.	NCT01800526 NCT01849016	[127]
Hemopexin	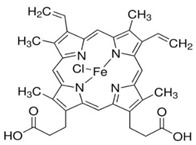	Decreases ROS by binding to free heme.	NCT06699849	[130]
Alpha-lipoic acid	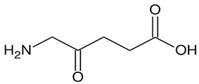	NRF2 agonist that mediates glutathione production.	NCT03161028	[131]
δ-aminolaevulinate	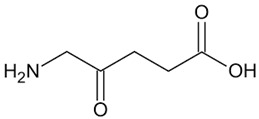	Metabolic precursor of protoporphyrin IX in heme biosynthesis.	NRF2 agonist and treatment modality for actinic keratosis.	[132]
L-Arginine	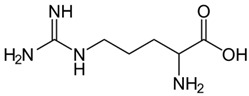	Precursor for nitic oxide.	Reduce ROS, improved mitochondrial function. NCT04839354	[107,133]
Sulforaphane	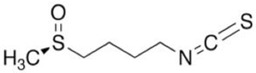	Increases NRF2 translocation to the nucleus.	NCT01715480	[133]

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
