# Peer review of "Innovations in Drug Discovery for Sickle Cell Disease Targeting Oxidative Stress and NRF2 Activation—A Short Review"

_ijms, 2025, doi:10.3390/ijms26094192_

Round 1

Reviewer 1 Report

Comments and Suggestions for Authors

The review article is very well written by the authors, clarifying ‘the oxidative stress’ SCD patients have to go through. US FDA has two approved treatments for SCD, namely L-glutamine and crizanlizumab that indirectly target the oxidative stress. SCD is a genetic disorder and the recently approved gene therapies, Casgevy and Lyfgenia hold tremendous potential in curing the patients. However, these therapies are costly, not easily accessible to SCD patients world wide and hence, there is always a need for new drugs to treat vaso-occlusive episodes associated with SCD. The pathophysiology associated with oxidative stress in SCD is well reviewed in this article and I agree with the authors that targeting oxidative stress through NRF2 activation represents a promising future for drug discovery in treating SCD. I highly recommend publishing this article in Int.J.Mol.Sci.

Below are 2 specific comments that need to be address:

  • Line 174, what does authors mean by unstable sickled RBCs?
  • Lines 173 to 181, rewrite, clarifying the cascade of events clearly for readers to understand properly.

Minor comment, line 212, n is missing from and.

Author Response

See attached response to reviewer file

Reviewer 2 Report

Comments and Suggestions for Authors

The authors described oxidative stress in SCD and ways of targeting it via NRF2 activation.

The review is interesting, well written and i have the following concerns:

  1. This review contains no figures. Please add 2 figures: one describing the pathophysiology of SCD and how to target oxidative stress in SCD via NRF2 activation and two, providing the possible drugs and how they might interact together or with other drugs in treating oxidative stress.
  2. Do the described relevant clinical trials in this review also include the use of HU? It is well known that HU is the gold-standard of treating SCD patients. Thus, many clinical trials include HU together with the novel drug under investigation.
  3. Are there any known side effects clinically by the drugs targeting oxidative stress in SCD via NRF2 activation?
  4. Could you provide some more information regarding the possible role of mitochondria axis and oxidative stress in SCD? Can this axis be targeted therapeutically?

Author Response

See attached response to reviewers' file

Round 2

Reviewer 2 Report

Comments and Suggestions for Authors

I have no further concerns.